# Sexual Behavior and Perceived Loneliness in Elderly People Living with HIV in China during the COVID-19 Pandemic

**DOI:** 10.3390/ijerph20032714

**Published:** 2023-02-03

**Authors:** Yushan Hou, Chang Cai, Houlin Tang, Yichen Jin, Fangfang Chen, Dandan Niu, Fan Lv

**Affiliations:** National Center for AIDS/STD Control and Prevention, Chinese Center for Disease Control and Prevention, Beijing 102206, China

**Keywords:** HIV/AIDS, loneliness, sexual needs, sexual behavior

## Abstract

Background: The proportion of elderly people living with HIV (PLHIV) is increasing in China. To advance targeted interventions and substantially improve their quality of life, we investigate indicators of loneliness and sexual behavior among elderly PLHIV in 10 districts/counties in China during the COVID-19 pandemic. Methods: The demographic information and laboratory test results of the potential respondents were initially collected from the China Information System for Disease Control and Prevention. A two-stage stratified cluster sampling was used. The questionnaire survey was individually provided to all PLHIV aged +60. Results: We recruited 1017 valid respondents with a median age of 66 years (interquartile range of 63–71), of which 776 (76.3%) were male. Overall, 341 respondents (33.5%) lived alone, and 304 (29.9%) felt lonely. A total of 726 respondents (71.4%) informed others of their HIV diagnosis. Among the 726 respondents, children were the most common group with whom the older people shared their HIV infection status, accounting for 82.9%. Approximately 20% of the older PLHIV engaged in sexual behavior in the last year, and 70% reported not using condoms. A significantly greater risk of loneliness was found among the females (AOR = 1.542, CI: 1.084, 2.193), those who suffered discrimination from informed people (AOR = 4.719, CI: 2.986, 7.459), were diagnosed <1 year prior (AOR = 2.061, CI: 1.345, 3.156), those living alone (AOR = 2.314, CI: 1.632, 3.280), those having no friends (AOR = 1.779, CI: 1.327, 2.386), and those who had a divorced or widowed marital status (AOR = 1.686, CI: 1.174, 2.421). Conclusions: Compared with non-lonely participants, the lonely participants were more likely to have a rural registered residence, a lower education level, no friends, be divorced or widowed, live alone, and lack knowledge of smartphones and reproductive health. The influence of COVID-19 had caused social activities to be more confined to the community, which impacts elderly HIV patients suffering from severe discrimination within families and communities.

## 1. Introduction

The HIV epidemic has brought major healthcare challenges due to the rise of cases in China. The data released by the National Center for AIDS/STD Control and Prevention demonstrated an alarming increase in new HIV diagnoses in the elderly population [1]. The annual number of newly reported HIV cases in people aged +60 increased by more than a factor of five between 2010 and 2018 [2]. Many challenges come with comprehending the health needs of older people living with the HIV (PLHIV). Their psychosocial needs are equally important and have garnered enormous attention from researchers. Loneliness is the most common concern regarding various psychosocial issues because it leads to depression, chronic diseases, and even death [3,4,5,6]. According to a nationwide longitudinal study on the elderly in China, 22.9% of the male and 30.6% of the female respondents reported suffering from loneliness [7,8]. The British Newspaper Association reported that 30.9% of respondents had experienced loneliness within one month after the outbreak; the survey included 50.8% of young people aged between 16 and 24, 33.2% of adults aged between 25 and 39, 26.2% aged between 40 and 54, 24.1% aged between 55 and 69, and 28.1% aged +70. In addition, 53% of people were alone (i.e., single, widowed, divorced, or separated alone) [9]. A cross-sectional study based on the rural elderly population in Shandong, China, found that 25.0% of respondents reported feeling lonely, with females accounting for 58.1% of the lonely respondents. The empty-nest elderly who were not living with spouses and were suffering from physical diseases were most susceptible to loneliness.

Aside from these components, older PLHIV indicated that suffering from HIV exacerbates their loneliness. Several reports indicate that the elderly PLHIV exhibit a higher prevalence of chronic diseases than their uninfected partners [10,11]. In addition, elderly people are more likely to live alone and possess a more limited social network compared to their younger partners [12]. The HIV-related stigma is a common barrier to building mutual emotional relationships. Due to public misconceptions about HIV transmission, outcomes, and contagiousness, newly infected older adults may fear being rejected by their loved ones due to the perceived stigma. This fear prevents them from reaching out, thereby increasing their feelings of loneliness [13]. Although some studies have reported loneliness in PLHIV, to the best of our knowledge, no study has explored the current sexual behavior and loneliness of older PLHIV during the COVID-19 pandemic.

This research gap may stem from the sexual needs of older people being considered taboo. Traditional Chinese gender norms also restrict women’s sexuality and practices, and minimal information exists about older women’s risk behaviors [14]. Evidence indicates that elderly people, specifically males, still experience strong sexual needs that are not fully perceived. After being widowed, divorced, or single for other reasons, elderly males often engage in extramarital sex or even commercial sex using condoms [15]. The lack of studies indicates that there is still room to explore the sexual needs and behaviors of HIV-infected older people. Accordingly, this study aimed to describe the prevalence and the risk factors of loneliness and HIV-related characteristics among elderly PLHIV. In doing so, we examined any associations with demographic factors, discrimination, and sexual behaviors. We also identified key correlates of loneliness, which can help reveal possible intervention targets to prevent or reduce loneliness and the associated adverse health consequences.

## 2. Methods

### 2.1. Two-Stage Stratified Cluster Sampling

Five provinces or municipalities in China were selected based on the geographical location and economic status. An economically developed central urban district and a developed county/district region from each province or municipality were then selected. All local PLHIV aged +60 were recruited from 10 survey sites. The contact information of the selected potential respondents was collected from the national HIV/AIDS case reporting system (CRS), which was established in 1985 and incorporated into a web-based version in 2005 [16]. All new HIV infections are reported through the CRS, and patients receive follow-up visits from local medical institutions and the Centers for Disease Control and Prevention (CDCs).

### 2.2. Data Collection and Management

The questionnaire survey was formulated by the National Center for AIDS/STD Control and Prevention. The investigation was conducted by systematically trained researchers. The ID number of the elderly PLHIV in the CRS was used as the identification mark. The local CDC staff, who followed up on the visits, subsequently contacted the elderly PLHIV and conducted the questionnaire interviews from April to May in 2021. The interviews were mostly conducted in person, with a telephone interview used for those who could not be reached.

The study had the following inclusion criteria: (1) the participants should be alive and must be aged ≥60 at the time of the first diagnosis by 31 March 2021; (2) the participants must have lived in the local area for at least six months, except in cases of mental disorders leading to an inability to communicate or PAI, and must provide informed consent; (3) the current sexual behavior and social habits (e.g., marital status, living situation, presence of friends, telephone use, and discrimination) of the subjects were collected [17].

### 2.3. Measures of Loneliness and HIV Disclosure Status and Discrimination

#### 2.3.1. Loneliness

Other studies have used different categories to measure loneliness. In our study, loneliness was measured from a single item (“I felt lonely”) in the CES-D 10 [18], which is in accordance with previous studies [19,20]. This measurement of loneliness has been proven to possess good face and predictive validity [21]. 

#### 2.3.2. HIV Disclosure Status and Discrimination

HIV disclosure and subsequent discrimination were further investigated. To explore this aspect, two questions were frequently asked: “did you inform others about your HIV infection?” and “did they react with discriminatory behaviors when they knew your HIV status?”. In this context, the discriminatory behavior refers to alienating or avoiding the participant; not being willing to dine with the participant; not being willing to use public facilities (e.g., toilets) with the participant; thoroughly cleaning items the participant touched; mocking, humiliating, or blaming the participant; using physical violence against the participant; or seriously emotionally harming the participant. Based on responses, the participants were dichotomized into three different groups (1 = “did not inform anyone”, 2 = “no discriminatory behavior from the informed people”, and 3 = “suffered discriminatory behavior from the informed people”).

### 2.4. Statistical Analysis

Demographics, sex-related characteristics, and behaviors, were described and compared based on gender. The continuous variables were summarized using the median and the interquartile range (IQR). Categorical variables were summarized using numbers and percentage values. The differences in the distribution of values were analyzed using the Chi-square test. Univariate and multivariate binary logistic regression models were used to assess determinants of loneliness with odd ratios (ORs) and 95% confidence intervals (CIs). All p-values were two-sided, considering *p* < 0.05 as statistically significant. All statistical analyses were performed using SPSS 22.0.

## 3. Results

### 3.1. Demographic Characteristics of Older PLHIV

The CRS initially recorded 1305 participants aged 60 at the time of diagnosis. Only 1017 valid respondents were recruited in our study. The remainder of the potential participants were not included in the study due to not following up or being unwilling or unable to answer the questionnaire. The median age of the respondents was around 66 years, with an IQR of 63–71. Among them, 776 respondents (76.3%) were male. There was no significant difference between lonely participants and non-lonely participants in age and gender. However, the comparison of the demographic characteristics of them showed that there were significant differences in registered residence, education, marital status, and living situation. (*p* < 0.05, Table 1).

### 3.2. HIV-Related Characteristics of Respondents

Among the subjects, 835 (82.1%) were infected with HIV from heterosexual contact. Following HIV infection diagnosis, 726 respondents (71.4%) informed others of their status. Among the people informed, children and spouse/sexual partner were the two most common groups, accounting for 82.9% and 57.9%, respectively, of those with whom the older people shared their HIV infection status. Among the 726 older PLHIV who shared their infection status, 142 (19.6%) reported experiencing discriminatory reactions from the people they informed.

Considering sexual orientations, a total of 863 (84.9%) participants identified as heterosexual. A total of 240 (23.6%) stated having sexual needs last year. Although approximately 20% were sexually active, 30.5% were willing to use condoms during every sexual encounter. Compared with the non-lonely participants, the lonely participants were more likely to become infected by heterosexual contact.

The lonely participants were not different from the non-lonely participants based on ART treatment and CD4 counts. There was a significant difference between lonely participants and non-lonely participants in the disclosure status and discrimination (*p* < 0.05, Table 2).

### 3.3. Risk Factor Analysis of Loneliness among Older People Living with HIV

Among the older PLHIV, 304 (29.9%) felt lonely, 464 (45.6%) had no feeling, and 249 (24.5%) were not lonely; 426 respondents were able to use a smart phone. Univariate analysis showed that being able to use a smart phone could reduce the risk of loneliness (OR = 0.727, CI: 0.551, 0.958), but it was not significant in the multivariate analysis. With a smart phone, 86.9% of them could chat and 62.2% could make payments. Talking about the number of friends who kept close contact, 184 (18.1%) respondents had more than 4 friends, 372 (36.6%) had 1–3 friends, and 461 (45.3%) had no friends. A significantly greater risk of loneliness was found among females (AOR = 1.542, CI: 1.084, 2.193), those who suffered discrimination from informed people (AOR = 4.719, CI: 2.986, 7.459), those who were diagnosed <1 year prior (AOR = 2.061, CI: 1.345, 3.156), those living alone (AOR = 2.314, CI: 1.632, 3.280), those having no friends (AOR = 1.779, CI: 1.327, 2.386), and those who had a divorced or widowed marital status (AOR = 1.686, CI: 1.174, 2.421) (Table 3).

## 4. Discussion

The increasing prevalence of HIV infection in elderly patients has become a new challenge in the AIDS epidemic in China [21]. The sexual behavior of older people is rarely thoroughly explored. It should be noted that older people, mainly males, have significant sexual urges. We found that nearly a quarter of the older male PLHIV indicated their sexual needs, which was lower than 52% in the general older male population from a recent study [22]. However, this is similar to that in other studies of older male PLHIV. This may be due to the role of psychology in older people and partly due to the ART medication. In our survey, we also found that 30.5% of the older PLHIV were willing to use condoms. The prime reason for unprotected sex could be not needing contraception. Older people intend to use condoms only for contraception, as they lack access to sex education and thus have insufficient AIDS knowledge. The elderly possess limited educational and comprehensive ability. This leads to the necessity of promotional materials, especially in rural areas. Adopting a variety of health publicity and education methods for publicizing AIDS-related knowledge in areas where the elderly gather is often required. Effective cautionary education should be provided to improve AIDS-related knowledge and risk awareness and promote the active use of condoms and early testing [23].

A recent study demonstrated the relationship between perceived discrimination and loneliness. Although discrimination was reported among medical staff or colleagues [24,25], the older PLHIV stated that greater discrimination came from neighbors, friends, and family members. The reason for suspicions of discrimination could be related to their living environment. As China is traditionally a family-centered culture, the one “safe harbor” for Chinese people is generally inside the family. The stigma within this core can destabilize the foundations of a person’s emotional and psychological well-being [26]. Unfortunately, the older PLHIV suffered more significant discrimination within their families. In our survey, the proportion of older PLHIV who disclosed their infection status to their families was lower than that in a previous study among adult patients [27]. This may be due to their fear of being discriminated and their reluctance to tell their families and friends. It should be noted that newly infected older adults might fear being rejected by their loved ones due to the perceived stigma. The perceived stigma could lead to shame and “silencing” or the non-disclosure of HIV status, which might increase feelings of loneliness.

The results of the multivariate analysis also demonstrated that in the first year of knowledge of the HIV infection, having no friends and living alone put them at risk of feeling lonely. We found that approximately 50% of the selected participants had no friends. Some participants (33.5%) were living alone; this percentage is significantly larger than that of elders in general [28]. In China, family members, particularly spouses and adult children, traditionally take care of older relatives. However, the disintegration of extended family members in recent years has contributed to the perception that adult children are unable to give older adults the unnecessary help or care. Although they seemed very lonely and many of them were uncertain about their feelings or had never thought about them, the older adults still do not ask their child to get help. This should inspire professionals to reach out to help them even more. In addition to timely free anti-viral therapy, they must have access to psychological counseling and humanitarian care. A previous study on elderly subjects indicated that despite receiving help from qualified individuals and living independently, the departure of adult children caused deep loneliness among older parents [29]. We should also encourage adult children to pay more attention to their parents.

The elderly have been lonely since the beginning of the COVID-19 pandemic. This loneliness is compounded by most of them having one-child families and by the children not being in their presence most of the time. In retirement, the elderly are gradually isolated from society. Social activities become infrequent, interpersonal connections become narrower, and the activity space shrinks. At the same time, the older PLHIV lack an opportunity for communication. COVID-19 has caused social activities to be more confined to the community. This causes suffering due to severe discrimination within communities [30].

### Limitations

This study has several limitations. First, many infected people died shortly after diagnosis, which caused survivor offset. Second, the social vulnerability remained unanswered. Third, this study used self-reported survey data, which may introduce information and recall bias. Future studies are needed to verify the findings of the current study.

## 5. Conclusions

Our findings have demonstrated a consistent need for interventions to reduce the HIV-related stigma, as well as services that prevent loneliness in the elderly PLHIV. Extensive research is needed to understand and identify the issues stemming from the multi-layered stigma in the elderly PLHIV. There must be appropriate intervention testing and assessments on whether stigma reduction intervention improves associated outcomes, which includes loneliness.

## Figures and Tables

**Table 1 ijerph-20-02714-t001:** Demographic characteristics of respondents.

Characteristics	Total (%)	Not Lonely (%)	Lonely (%)	*p*-Value
Median age (IQR)	66 (63–71)	66 (63–71)	66 (63–70)	
60–69	704 (69.2)	505 (70.8)	199 (65.5)	0.090
≥70	313 (30.8)	208 (29.2)	105 (34.5)	
Gender				0.077
Male	776 (76.3)	555 (77.8)	221 (72.7)	
Female	241 (23.7)	158 (22.2)	83 (27.3)	
Registered residence				0.002
Rural	564 (55.5)	373 (52.3)	191 (62.8)	
Urban	453 (44.5)	340 (47.7)	113 (37.2)	
Educational level				0.002
Primary school or illiterate	434 (42.7)	292 (41.0)	142 (46.7)	
Middle school	500 (49.2)	349 (48.9)	151 (49.7)	
College and above	83 (8.2)	72 (10.1)	11 (3.6)	
Marital status				0.000
Unmarried	64 (6.3)	44 (6.2)	20 (6.6)	
Married or remarried	590 (58.0)	459 (64.4)	131 (43.1)	
Divorced or widowed	363 (35.7)	210 (29.5)	153 (50.3)	
Living situation				0.000
Alone	341 (33.5)	187 (26.2)	154 (50.7)	
Others	676 (66.5)	526 (73.8)	150 (49.3)	
Have friends				0.000
Yes	556 (54.7)	418 (58.6)	138 (45.4)	
No	461 (45.3)	295 (41.4)	166 (54.6)	
Using smartphone				0.023
Yes	426 (41.9)	315 (44.2)	111 (36.5)	
No	591 (58.1)	398 (55.8)	193 (63.5)	

**Table 2 ijerph-20-02714-t002:** HIV-related characteristics of respondents.

Characteristics	Total	Not Lonely	Lonely	*p*-Value
	N (%)	N (%)	N (%)	
Infection routes				0.004
Homosexual contact	166 (16.3)	134 (18.8)	32 (10.5)	
Heterosexual contact	835 (82.1)	567 (79.5)	268 (88.2)	
Others	16 (1.6)	12 (1.7)	4 (1.3)	
Years since HIV diagnosis				0.001
<1	153 (15.0)	91 (12.8)	62 (20.4)	
1–5	457 (44.9)	315 (44.2)	142 (46.7)	
>5	407 (40.0)	307 (43.1)	100 (32.9)	
Disclosure status and discrimination				0.000
Not informing anyone	291 (28.6)	215 (30.2)	76 (25.0)	
No discriminatory behavior	584 (57.4)	439 (61.6)	145 (47.7)	
Suffering discriminatory behavior	142 (14.0)	59 (8.3)	83 (27.3)	
On ART				0.267
Yes	944 (92.8)	666 (93.4)	278 (91.4)	
No	73 (7.2)	47 (6.6)	26 (8.6)	
CD4 counts (cells per mm^3^)				0.312
<200	158 (15.9)	106 (15.1)	52 (17.7)	
≥200	837 (84.1)	595 (84.9)	242 (82.3)	
Sexual orientation				0.037
Homosexuality	101 (9.9)	81 (11.4)	20 (6.6)	
Heterosexuality	863 (84.9)	592 (83.0)	271 (89.1)	
Bisexuality or other	53 (5.2)	40 (5.6)	13 (4.3)	
Sexual needs				0.599
Yes	240 (23.6)	165 (23.1)	75 (24.7)	
No	777 (76.4)	548 (76.9)	229 (75.3)	
Number of sexual partners				0.000
Two or more	34 (3.3)	17 (2.4)	17 (5.6)	
One	163 (16.0)	132 (18.5)	31 (10.2)	
None	820 (80.6)	564 (79.1)	256 (84.2)	

**Table 3 ijerph-20-02714-t003:** Risk factors of loneliness.

Characteristics	Feeling Lonely N (%)	Unadjusted OR (95% CI)	*p*	Adjusted OR (95% CI)	*p*-Value
Gender					
Male	221 (28.5)	1		1	
Female	83 (34.4)	1.319	0.078	1.542 (1.084, 2.193)	0.016
Discrimination from informed people					
Did not inform anyone	76 (26.1)	1		1	
No discriminatory behavior	145 (24.8)	0.934	0.680	1.038 (0.734, 1.468)	0.834
Suffered discriminatory behavior	83 (58.5)	3.980	0.000	4.719 (2.986, 7.459)	0.000
Years since HIV diagnosis					
<1	62 (40.5)	2.092	0.000	2.061 (1.345, 3.156)	0.001
1–5	142 (31.1)	1.384	0.034	1.304 (0.938, 1.812)	0.114
>5	100 (24.6)	1		1	
Living alone					
No	150 (22.2)	1		1	
Yes	154 (45.2)	2.888	0.000	2.314 (1.632, 3280)	0.000
Have friends					
Yes	138 (24.8)	1		1	
No	166 (36.0)	1.704	0.000	1.779 (1.327, 2.386)	0.000
Marital status					
Married or remarried	131 (22.2)	1		1	
Unmarried	20 (31.3)	1.593	0.105	1.056 (0.553, 2.019)	0.868
Divorced or widowed	153 (42.1)	2.553	0.000	1.686 (1.174, 2.421)	0.005

## Data Availability

The data are available upon reasonable request.

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
