# Peer review of "Sexual Behavior and Perceived Loneliness in Elderly People Living with HIV in China during the COVID-19 Pandemic"

_ijerph, 2023, doi:10.3390/ijerph20032714_

Round 1

Reviewer 1 Report

Please, I suggest that the authors review the methods and results sections. There are texts that do not agree with the objectives and methodology.

Reviewer 2 Report

How was the reliability of the "loneliness" evaluated in your study?

How was the validity and reliability of the “HIV disclosure status and discrimination” evaluated?

In the results section, when you have fully mentioned the findings in the tables, there is no need to repeat them in full detail in the text. Please summarize the findings in the text.

The discussion section needs serious revision. In the discussion section, you have repeated the results, which is better to justify and criticize the results of your study with others.
